# Dextromethorphan Reduces Oxidative Stress and Inhibits Uremic Artery Calcification

**DOI:** 10.3390/ijms222212277

**Published:** 2021-11-13

**Authors:** En-Shao Liu, Nai-Ching Chen, Tzu-Ming Jao, Chien-Liang Chen

**Affiliations:** 1Division of Cardiology, Kaohsiung Veterans General Hospital, Kaohsiung 813779, Taiwan; esliou@vghks.gov.tw; 2Institutes of Clinical Medicine, National Yang Ming Chiao Tung University, Hsinchu City 30010, Taiwan; 3Department of Neurology, Kaohsiung Chang Gung Memorial Hospital, Chang Gung University College of Medicine, Kaohsiung 813779, Taiwan; naiging@yahoo.com.tw; 4Divisions of Medical Research, Kaohsiung Veterans General Hospital, Kaohsiung 813779, Taiwan; tmjao@vghks.gov.tw; 5Institution of Precision Medicine, National Sun Yat-sen University, Kaohsiung 813779, Taiwan; 6Divisions of Nephrology, Kaohsiung Veterans General Hospital, Kaohsiung 813779, Taiwan; 7Faculty of Medicine, School of Medicine, National Yang Ming Chiao Tung University, Hsinchu City 30010, Taiwan

**Keywords:** dextromethorphan, smooth muscle cell, arterial calcification

## Abstract

Medial vascular calcification has emerged as a key factor contributing to cardiovascular mortality in patients with chronic kidney disease (CKD). Vascular smooth muscle cells (VSMCs) with osteogenic transdifferentiation play a role in vascular calcification. Nicotinamide adenine dinucleotide phosphate (NADPH) oxidase inhibitors reduce reactive oxygen species (ROS) production and calcified-medium–induced calcification of VSMCs. This study investigates the effects of dextromethorphan (DXM), an NADPH oxidase inhibitor, on vascular calcification. We used in vitro and in vivo studies to evaluate the effect of DXM on artery changes in the presence of hyperphosphatemia. The anti-vascular calcification effect of DXM was tested in adenine-fed Wistar rats. High-phosphate medium induced ROS production and calcification of VSMCs. DXM significantly attenuated the increase in ROS production, the decrease in ATP, and mitochondria membrane potential during the calcified-medium–induced VSMC calcification process (*p* < 0.05). The protective effect of DXM in calcified-medium–induced VSMC calcification was not further increased by NADPH oxidase inhibitors, indicating that NADPH oxidase mediates the effect of DXM. Furthermore, DXM decreased aortic calcification in Wistar rats with CKD. Our results suggest that treatment with DXM can attenuate vascular oxidative stress and ameliorate vascular calcification.

## 1. Introduction

Cardiovascular disease (CVD) is a leading cause of mortality and morbidity among patients with chronic kidney disease (CKD) undergoing chronic dialysis [1]. According to the United States Renal Data System, CVD accounts for approximately 40% of mortality among patients on dialysis and is the main cause of hospitalization [2]. Traditional factors such as hypertension, smoking, hyperlipidemia, and non-traditional risk factors, such as anemia, abnormal calcium/phosphorus metabolism, hyperhomocysteinemia, extracellular fluid volume overload, oxidative stress, malnutrition, and inflammation, have been implicated in CVD development in patients on chronic dialysis [3].

Arterial calcification is much more prevalent in patients with CKD or chronic dialysis than in those without kidney disease and contributes to extremely high morbidity and mortality rates [3,4,5]. Vascular calcification consists mainly of arteriosclerosis (medical calcification) with vascular stiffness and atherosclerosis (intimal calcification) with narrowing of the vessel lumen. Arteriosclerosis and atherosclerosis may coexist in dialysis patients with CVDs. The pathogenesis of vascular calcification is a complex multifactorial process attributed to inflammation, metabolic disorders, and genetic causes [6,7,8]. It is widely accepted that vascular calcification is a highly regulated cell-mediated process that resembles bone formation in many aspects. Artery calcification inducers and inhibitors of such phenotype switching have been identified, and alterations in the balance of these pro- and anti-calcific stimuli are considered to eventually cause ectopic mineral deposition [8]. Vascular smooth muscle cells (VSMCs) present in the tunica media play a central role in vascular calcification. VSMCs can undergo osteochondrogenic transdifferentiation, resulting in a cell-type resembling osteoblasts or chondrocytes.

Reactive oxygen species (ROS) are byproducts of aerobic metabolism. Several enzymes in the body can produce ROS. Nicotinamide adenine dinucleotide phosphate (NADPH) oxidase is a major source of ROS in the cardiovascular system and plays a major role in mediating redox signaling under pathological conditions. Oxidative stress and excessive ROS production are important mediators of osteochondrogenic transdifferentiation in VSMCs [8,9,10]. Hyperphosphatemia is known to induce ROS production in patients with CKD and in animals [11,12,13,14]. The molecules used for ROS removal or decrease may thus provide new targets for the cardiovascular system. NADPH oxidase inhibitors, such as diphenyleneiodonium chloride and apocynin, reduce ROS production and block calcified-medium–induced VSMC calcification [15,16].

Dextromethorphan (DXM), an *N*-methyl-d-aspartate (NMDA) receptor agonist, is the dextrorotatory isomer of the codeine analog levorphanol, a morphinan. The NMDA receptor expressed on rat vascular smooth muscle (A7r5) cells determines the effects of homocysteine in dysregulating vascular factors that modulate cell proliferation and migration [17]. DXM is also an NADPH oxidase antagonist [18,19,20] and has been shown to improve blood pressure control and dilate blood vessels [20,21]. Thus, we hypothesized that DXM could attenuate vascular smooth muscle cell transdifferentiation to osteogenic cells in vitro and vascular calcification in vivo in animal models of CKD.

## 2. Results

### 2.1. Effect of High-Phosphate Medium on Rat Vascular Smooth Muscle Cells

Diffuse calcification can be induced by beta-glycerophosphate (β-GP) calcification medium. The calcification model is useful for analyzing the molecular and cellular mechanisms of vascular calcification. The high-phosphate medium induced granular deposits in the cell layer with cellular phenotypes within a few days (Figure 1A). These deposits coalesced into nodules, finally resulting in diffuse calcification of the cell layer by day 14. DXM decreased granular deposits in a dose-dependent manner. However, DXM did not induce proliferation or overt death of vascular smooth muscle cells (Figure 1B). A growing body of evidence suggests the contribution of ROS involvement and redox signaling in the osteochondrogenic transdifferentiation of VSMCs. DXM significantly decreased the amount of calcified-medium–related ROS in a dose-dependent manner (*p* < 0.05; Figure 1C). High phosphate medium caused a decrease in ATP and mitochondria membrane potential in A7r5 cells and this effect was ameliorated by DXM at the concentration of 50 μM (Figure 1D,E). Taken together, the results show that DXM combated the oxidative stress induced by high phosphate medium and produced healthier mitochondria.

#### Effect of DXM on High-Phosphate Medium in Rat Smooth Vascular Cells

Alizarin red staining revealed a significant decrease in calcification by DXM in a dose-dependent manner (Figure 2a). However, the addition of an NMDA receptor antagonist (dizocilpine, MK-801, Sigma-Aldrich) did not alter calcification. This suggests that DXM inhibits osteogenic transdifferentiation in vascular smooth muscle independent of the NMDA antagonist (Figure 2a,b).

### 2.2. Effect of High-Phosphate Medium on Human Aortic Vascular Smooth Muscle Cells

As shown in Figure 3A,B, high-phosphate medium (CaCl_2_, NaH_2_PO_4_, and Na_2_HPO_4_) decreased cell viability and increased calcification transformation in human aortic vascular smooth muscle cells (HASMCs). DXM significantly decreased calcification in HASMCs in high-phosphate medium (*p* < 0.01). NADPH oxidase inhibitors such as apocynin can inhibit NADPH oxidase and block calcified-medium–induced VSMC calcification. In this experiment, the protective effect of DXM was not further increased by treatment with high-dose apocynin, indicating that NADPH oxidase is a major mediator of the effect of DXM. However, we cannot completely exclude other minor factors related to the VSMC phenotype and calcification.

### 2.3. Effects of DXM on Adenine Rat Models

To determine whether DXM lowers arterial calcification levels, we used an experimental rat model of adenine-induced renal failure with hyperphosphatemia. Rats were fed a 0.75% adenine diet to induce CKD. DXM was administered orally from the first 21 days after adenine feeding until the animals were sacrificed on day 43 (Figure 4A). The adenine diet retarded the increase in body weight of rats compared to a non-adenine diet (*p* < 0.01; Figure 4B). After stopping the adenine diet, the rats showed an increase in their body weight. Despite recovery, body weight was insufficient for complete recovery. All rats that were fed adenine were characterized by enlarged kidneys. Adenine-fed rats had enlarged kidneys with dark-stained material diffusely distributed throughout the cortical region, as observed via hematoxylin–eosin staining and von Kossa staining (Figure 4C). Ectopic calcification was histopathologically observed in the renal tubules and in the tubular basement membrane in the adenine control group, but not in the normal control group (Figure 4C).

Furthermore, the rats on an adenine diet showed a significant increase in systolic, rather than diastolic, blood pressure at days 21 and 42 (Figure 5A,B; *p* < 0.001 and *p* > 0.05, respectively). Rats on an adenine diet showed chronic renal failure with evidence of increased serum blood urea nitrogen (BUN; Figure 5C), serum creatinine (CREA; Figure 5D), and phosphate levels (Figure 5E). There were no significant differences in systolic blood pressure; diastolic blood pressure; and serum BUN, CREA, and phosphate levels between the rats fed the adenine diet with and without DXM. DXM did not significantly improve renal function in terms of serum CREA or serum BUN levels.

### 2.4. DXM Reduces Vascular Calcification in a Rat Model of CKD with Hyperphosphatemia

To examine the efficacy of DXM in preventing ectopic calcification in soft tissues, the calcium and phosphorus levels of the thoracic aorta were analyzed by hematoxylin-and-eosin staining and von Kossa staining. Figure 6A shows that there was no vascular calcification in Group 1 (the normal control rats without renal failure). Histological assessment using von Kossa staining showed that rats in Group 2 (adenine diet without DXM) had more extensive medial artery calcification (arteriosclerotic disease) than rats in Group 1 (without adenine diet) (Figure 6A,B). There was no calcification in the intimal layers of the arterial wall (atherosclerotic artery disease) in rats fed the adenine diet without DXM. However, ectopic calcification was significantly reduced in the regions with calcification for rats fed the adenine diet with DXM (Group 3) compared with those fed the adenine diet without DXM (Figure 6A,B).

As shown in Figure 6A, there was a significant downregulation of runt-related transcription factor 2 (RUNX2) expression in Group 1 (normal control rats without renal failure). RUNX2 expression was upregulated in the regions with ectopic calcification in Group 2 (adenine diet) rats. However, RUNX2 expression was significantly downregulated in the regions with ectopic calcification in rats fed the adenine diet with DXM (Group 3) compared to Group 2 (adenine diet without DXM; Figure 6A,C).

In summary, DXM reduced vascular calcification in a rat model of CKD with hyperphosphatemia (Figure 6A–C).

A schematic diagram (Figure 7) illustrates the renal failure associated with the downstream superoxide machinery, which contributes to vascular calcification. Thus, blockade of superoxide by DXM could decrease vascular calcification.

## 3. Discussion

The present study demonstrated that DXM treatment could inhibit VSMC–osteoblast transdifferentiation and superoxide production. Furthermore, we showed that treatment with DXM decreases aortic medial calcification in adenine renal-failure animal models. Our findings warrant further investigation into the potential therapeutic use of DXM for the reduction of chronic renal failure-related CVD.

Several studies have revealed that oxidative stress could lead to the dysfunction of various organs, including CVD in CKD [22,23]. The endothelium is an important target in the pathogenesis of CVD in patients with CKD and hyperphosphatemia. Moreover, hyperphosphatemia impairs endothelial function by increasing ROS, inhibiting endothelial nitric oxide synthase, increasing oxidative stress, and inducing apoptosis in endothelial cells [11,12,13,14]. Previous studies have shown that calcifying VSMCs treated with inorganic phosphate exhibit mitochondrial dysfunction, as demonstrated by decreased mitochondrial membrane potential and ATP production, disruption of mitochondrial structural integrity, and concurrently increased ROS production [11,12,19,20,21,22,23]. Our study showed that a calcified medium could induce ROS. Furthermore, in our study, DXM attenuated mineralization of both human and rat VSMCs, across two types of high-phosphate media (Figure 1, Figure 2 and Figure 3) [11,12]. These results suggest potential clinical applications.

DXM has been reported to be neuroprotective against glutamate excitatory toxicity and degeneration of dopaminergic neurons through antagonization of the NMDA receptor. In this study, we found that NMDA receptor antagonists did not inhibit arterial calcification. We suggest that DXM decreases arterial calcification independently of NMDA receptors. Hyperphosphatemia is known to induce ROS production (Figure 1C). Our results confirm these observations in response to high phosphate medium-induced mitochondrial dysfunctions such as ROS production, ATP depletion, and MMP reduction in vascular smooth muscle cells s and demonstrate that DXM could ameliorate these effects (Figure 1C–E). Oxidative stress and excessive ROS production are key mediators of osteochondrogenic transdifferentiation in VSMCs [9,10]. Intravascular ROS can theoretically be produced by many enzymes, including xanthine oxidoreductase, uncoupled nitric oxide synthase, and NADPH oxidase [24,25,26,27,28]. NADPH oxidase is a major source of ROS in the cardiovascular system and plays a major role in mediating redox signaling under pathological conditions. NADPH oxidase is the target of DXM action because the DXM-mediated effect disappears in NADPH oxidase-deficient mice [18]. NADPH oxidase inhibitors, such as apocynin, reduce ROS production and block calcified-medium–induced VSMC calcification [15,16]. In this experiment (Figure 3B), although high-dose apocynin did not further increase the protective effect of DXM—highlighting the, at least partial, role of NADPH oxidase in mediating the effect of DXM. However, the effect of other minor factors related to the VSMC phenotype and calcification cannot be excluded.

Evidence has demonstrated that arterial calcification is an active, cell-regulated process, based on the discovery that vascular smooth muscle cell populations are responsible for maintaining proper vascular tone and can undergo transdifferentiation into osteoblast-like cells, resulting in increased vascular stiffness [4,5,6,7,8]. Our report showed that calcification was significantly reduced in the aortic area when DXM was added. The effects of DXM on the osteochondrogenic differentiation markers, such as Runx2 [29], were also decreased, consistent with the von Kossa staining vascular calcification results. Elevated serum phosphorus levels are known to promote vascular calcification in patients with CKD [11,12,13,14,30]. In this study, the phosphorus, urea, and CREA levels were elevated in rats, but treatment with DXM had no significant impact on renal function in rats with adenine-induced renal failure. There were no significant changes in blood pressure between rats fed the adenine diet with and without DXM, although DXM improved endothelial function, decreased blood pressure [20,21], and reduced the thickness of the medial layer of the aorta in hypertensive rats [19]. This discrepancy between the lack of change in blood pressure and its antioxidative effects on NADPH oxidase is difficult to explain. The possibly cause is the DXM dosages. The effect of DXM on blood pressure had been studied and the result showed that such effects are not dose-dependent. A low dose rather than a high dose of DXM could reduce blood pressure in experimental hypertension [20]. Extremely higher dose of DXM as the NMDA antagonist produces serotonergic-glutamatergic interactions in the mechanism of action of classic hallucinogens and increased blood pressure and heart rate [31]. Volunteers with extreme high doses (100 to 800 mg of kg) dextromethorphan, an NMDA antagonist, increase slighter higher blood pressure (6–17 mmHg) [31]. Hence, the serotonergic–glutamatergic interaction may offset the beneficial effects of DXM, such as lowering blood pressure. Daily oral DXM 20 mg/kg (Sigma-Aldrich) may be too higher for rats with CKD in concern of blood pressure. Future investigations are required to define the optimal dose of DXM before it could be used for another treatment dose in animal with CKD and patients with CKD [20]. Taken together, this study indicates that the action of DXM against vascular calcification is independent of renal failure improvement and mechanical stretch due to blood pressure changes. The anti-vascular calcification of DXM mostly depends on its direct action on local VSMCs because of its antioxidative effects.

DXM, an effective over-the-counter antitussive agent, is one of the most widely used cough-suppressant active ingredients in cold and cough medications, with high safety and efficacy at recommended doses. DXM is rapidly absorbed when administered orally, and is excreted mostly through the urine. It has largely replaced codeine as a cough suppressant. Unlike codeine, DXM is devoid of analgesic properties, and produces less respiratory depression, less gastrointestinal disturbance, and less drug dependence or abuse. DXM has been safely administered orally at 10–40 mg/kg in mice [19]. DXM reduces oxidative stress and inhibits typical inflammatory diseases involving macrophage proliferation and calcification in the intimal artery (atherosclerosis) in mice [19]. Addiction does not usually occur even after large doses for prolonged periods [19,29,32]. At equipotent doses for local anesthesia, DXM—compared with bupivacaine, a long-acting local anesthetic—was found to be lower in cardiovascular toxicity and safer for the central nervous system. The highest subcutaneous injected dose of DXM was 20 µmol/kg [33]. Accordingly, in this study, DXM was administered orally to rats at daily doses of 20 mg/kg to induce vascular calcification, including calcification in the intimal artery (atherosclerosis) and medial artery (arteriosclerosis). DXM has an excellent clinical safety record because it has been widely used as an over-the-counter anti-cough agent for several decades. However, volunteers with extreme high doses (100, 200, 300, 400, 500, 600, 700, 800 mg of kg) dextromethorphan, produce effects similar to classic hallucinogens and slighter higher blood pressure (6–17 mmHg) without life threatening events [31]. Owing to its proven safety record of long-term clinical use in humans, DXM may offer a therapeutic strategy for targeting vascular calcification and atherosclerosis in CKD. Therefore, there is a strong rationale for further studies.

The present study has some limitations that need to be addressed. First, the results of our study could not be directly extrapolated to humans, as data on other animal models are limited; therefore, further investigations are needed in other models or humans. However, the findings of the current study provide evidence for the involvement of oxidative stress in the mechanism underlying vascular calcification and indicate the potential for DXM as a clinical drug for preventing vascular calcification in patients with CKD. Second, this study confirmed the effects of dextromethorphan (DXM), on oxidative stress and vascular calcification. Actually, artery calcification is a far more complex setting of a blood vessel (including endothelial cells, adventitial cells, and so on). We only showed the vascular calcification evidence by hematoxylin–eosin staining, von Kossa staining and osteogenic differentiation of VSMC by RUNX2 in animal studies. DXM significantly attenuated the increase in ROS production, the decrease in ATP and mitochondria membrane potential during calcified-medium–induced VSMC calcification process (*p* < 0.05). Further detail mechanism needs further studies. However, we cannot completely exclude other signaling pathway effects.

## 4. Materials and Methods

### 4.1. Rat Vascular Smooth Muscle Cell Culture and Osteoblast Differentiation

A commercially available rat (*Rattus norvegicus*) aortic vascular smooth muscle cell (VSMC) line, A7r5, was purchased from ATCC via the Bioresource Collection and Research Center of the Food Industry Research and Development Institute, Hsinchu City, Taiwan. A7r5 cells were cultured in basal growth medium (DMEM, 10% FBS, 1% penicillin/streptomycin) at 37 °C with 5% carbon dioxide. To induce osteogenic differentiation, cells were first cultured in basal growth medium until 70% confluence and were then transferred to osteogenic medium in the presence of 10 mmol/L β-GP calcification medium for 14 days [11]. VSMCs were incubated with DXM (0, 10, and 50 µM; Sigma, St Louis, MO, USA) in the presence or absence of the NMDA receptor antagonist MK-801 (50 µM, Sigma, St Louis, MO, USA). The calcification medium supplement was applied simultaneously with various treatments (control; DXM: 0.5, 5, and 50 μM).

#### 4.1.1. MTT (3-[4,5-Dimethylthiazol-2-yl]-2,5 Diphenyl Tetrazolium Bromide) Assay

Rat VSMCs were seeded at a density of 5 × 10^3^ cells/well in a 96-well plate and allowed to attach overnight. Subsequently, the standard culture medium (DMEM, 10% FBS, antibiotics) was replaced with calcification medium supplemented with various treatments (control, and DXM 0.5, 5, and 50 μM). Cell proliferation was assessed on days 1–3 using a Cell Proliferation Kit II (XTT, Roche, Mannheim, Germany), a colorimetric assay for the non-radioactive quantification of cell proliferation and viability.

#### 4.1.2. Measurement of ROS Levels

ROS levels were detected with the ROS assay kit (Abcam, ab113851) according to the manufacture’s instruction. Moreover, 2′,7′-dichlorofluoroscein is highly fluorescent and the fluorescence is detected by fluorescence spectroscopy with excitation/emission at 485 nm/535 nm to represent the ROS levels.

#### 4.1.3. Measurement of Mitochondria ATP Generation

BioTracker ATP-Red Live Cell Dye for cellular adenosine triphosphate (ATP) localized to mitochondria is to detect cell health and metabolic activity. Briefly, A7r5 cells (1 × 10^5^ cells/mL) were incubated with various concentrations of different groups (control as DMEM alone) with and without high phosphate medium for 24 h. After treatment for 24 h, the cells were harvested with trypsin, washed with PBS, and resuspended in 200 ng/mL of ATP-Red Live Cell Dye (Sigma-Aldrich SCT045). After incubation for 30 min at 37 °C, the cells were washed thrice by PBS. Then, cells were immediately analyzed by fluorescence quantification by flow cytometry.

#### 4.1.4. Mitochondrial Membrane Potential

Rhodamine 123 (Invitrogen, Life Technologies, Carlsbad, CA, USA) was used to measure mitochondrial membrane potential. Briefly, A7r5 cells (1 × 10^5^ cells/mL) were incubated with various concentrations of different groups with and without high phosphate medium for 24 h. After treatment for 48 h, the cells were harvested with trypsin, washed with PBS, and resuspended in 200 ng/mL of Rhodamine 123. After incubation for 30 min at 37 °C, the cells were washed thrice and resuspended in 500 mL of PBS. After being washed with PBS, cells were immediately analyzed by flow cytometry.

#### 4.1.5. Detection of Mineralization

A7r5 cells were grown to subconfluence in 6-well dishes, then placed in serum-reduced medium and treated. The cells were then rinsed with water, drained, and stained with 2% alizarin red solution (pH 6.0). After 30 s incubation at room temperature, the plates were rinsed three times with distilled water. Alizarin red S (Sigma, St. Louis, MO, USA) staining was used to assess Ca deposition in VSMC cell layers, as Alizarin red S dye binds Ca ions in the cell layer matrix. The culture plates were photographed under a light microscope and assessed for mineralized nodules-stained red.

### 4.2. Primary Human Aorta Vascular Smooth Muscle Cell Culture and Osteoblast Differentiation

Primary HASMCs were purchased from Innoprot (Derio, Spain). Primary HASMCs were cultured as previously described [12]. Once the cells were confluent, they were seeded at a density of 1.5 × 10^4^ cells/well (96-well plate) for 24 h. The experimental medium consisted of culture medium supplemented with DMEM adding CaCl_2_, NaH_2_PO_4_, and Na_2_HPO_4_ to reach a final concentration of 2.5 mM inorganic phosphate and 2 mM calcium in the medium for 72 h. The deposition of calcium-phosphate crystals was assessed by alizarin red staining, as described previously [12]. The cells were then washed and stained with 1% alizarin red (Sigma-Aldrich, TMS-008, St. Louis, MO, USA), which chelates calcium to form a red precipitate. To quantify the amount of precipitate formed, cells were dissolved in 10% acetic acid and the alizarin red absorption was measured at 450 nm using a plate spectrophotometer. The calcification medium supplement was applied simultaneously with various treatments (control; DXM: 0.5, 5, and 50 μM; apocynin: 500 μM).

### 4.3. Animal Model

A vascular calcification model with chronic renal failure was established as previously described [25]. All animal experimental protocols used for treating the rats were approved by the Institutional Animal Care and Use Committee of the Kaohsiung Veterans General Hospital and conformed to the Guide for the Care and Use of Laboratory Animals (Kaohsiung Veterans General Hospital code: vghks-2017-2018-A057). Sixteen-week-old male Wistar Kyoto rats were obtained from the National Science Council Animal Facility (Taipei, Taiwan) and housed in the animal room of Kaohsiung Veterans General Hospital (Kaohsiung, Taiwan). The rats were kept in individual cages in a light-controlled room (12 h light/12 h dark cycle), and the temperature was maintained between 23 °C and 24 °C. The animals were randomly assigned to three groups: Group 1 (the control group with a normal diet), Group 2 (the CKD model group with a diet containing 0.75% adenine and 0.9% phosphorus), and Group 3 (the CKD model with a diet containing 0.75% adenine and 0.9% phosphorus, followed by daily oral DXM 20 mg/kg; Sigma-Aldrich).

#### 4.3.1. Non-Invasive Blood Pressure Measurement

Systolic and diastolic blood pressures were measured using the tail-cuff method (CODA, Kent Scientific, Torrington, CT, USA), which uses volume–pressure recording to detect blood pressure based on volume changes in the tail, as previously described [34,35]. The operating procedures were performed according to the manufacturer’s instructions. Briefly, the rats were trained in a rodent holder and placed on a warm plate (35 °C) three times before the beginning of the experiment. Measurements were taken 15 times and restricted to 40 min and below for the entire procedure; systolic blood pressure was recorded at the end of the experiment. To prevent blood pressure fluctuation resulting from circadian rhythms, the data were collected between 09:00 and 12:00 am, coordinated universal time with +08:00 offset for the Taiwan time zone.

#### 4.3.2. Biochemical Analysis

Blood samples were obtained from the rats at baseline and then at three-week intervals until the end of the study. Serum urea and CREA levels were measured. After 42 days of renal failure, heart blood samples were collected after the rats were euthanized via excess CO_2_ gas. The serum was separated via centrifugation and frozen for subsequent analysis. The aorta, including the thoracic and abdominal aorta, was immediately harvested, fixed in 10% formalin or snap-frozen in liquid nitrogen, and stored at −80 °C until further examination. The aortic tree was perfused with 20 mL phosphate-buffered saline using a 26 G cannula inserted into the left ventricle, allowing unrestricted reflux from an incision into the right atrium. The uremic rat model was confirmed via increased levels of serum creatinine, serum BUN, and blood phosphorus. Serum levels of phosphorus, total calcium, BUN, and CREA were measured using a quantitative colorimetric assay kit (Sigma, St. Louis, MO, USA) and an AutoAnalyzer (OLYMPUS, AU2700, Olympus CO Ltd., Tokyo, Japan). Serum Ca levels were measured using commercial kits. (Sigma-Aldrich, St. Louis, MO, USA).

#### 4.3.3. Histological Assessment

Immunohistochemical analysis of the rat thoracic aorta sample sections was carried out as described in a previous study [30]. Formalin-fixed paraffin sections of thoracic aortas were cut into 5-μm thick subsections, deparaffinized, and then rehydrated. To detect calcification in the aortas, cross-sections from each group were subjected to hematoxylin–eosin staining to demonstrate the extent of mineralization. For von Kossa staining, the tissue sections were deparaffinized and rehydrated before staining for arterial calcification. Briefly, the sections were incubated in 5% silver nitrate for 1 h, exposed to light using a 100 W lamp, washed three times in distilled water, and then placed in 5% sodium thiosulfate for 5 min. They were washed twice in water and soaked in nuclear fast red for 2 min. Finally, the tissues were washed three times with distilled water and then dehydrated. The extent of atherosclerosis was expressed as the percentage of the entire aortic surface area covered by the lesions. Quality assurance tests and confirmation of diagnoses were performed by staining the paraffin-embedded block sections with hematoxylin and eosin. The expression RUNX2, a master regulator of bone development, was determined using immunohistochemistry (anti-RUNX2 antibody, 1:100, sc-101145; Santa Cruz Biotechnology). The extent of atherosclerosis was expressed as the percentage of RUNX2-positive cell of the entire aorta. Five photos were taken for each sample in a clockwise direction, and the image was captured on a microcomputer and analyzed by a computerized toolbox using *Image J* analysis (ImageJ 2.0.0/1.53c/Java 1.8.0_172 (64-bit).

### 4.4. Data Analysis

Data are presented as the mean ± SEM. Group differences were examined using GraphPad Prism5 (GraphPad Software Inc., San Diego, CA, USA). Statistical significance was determined using comparisons among independent groups of data, and the independent *t*-test or one-way ANOVA, with Bonferroni post-test, were used to determine the presence of significant differences.

## 5. Conclusions

In the present study, we found that DXM inhibited chronic renal failure–induced calcification by preventing oxidative stress. The effect of DXM was dependent on the dose used for treatment or administration and on the restoration of the antioxidant pathway, which prevented VSMC-associated phenotypic changes. To date, no therapeutic intervention exists to specifically target vascular calcification. Therefore, rebalancing the oxidative state of the vasculature using DXM may be beneficial in preventing osteochondrogenic transdifferentiation of VSMCs and subsequent vascular calcification.

## Figures and Tables

**Figure 1 ijms-22-12277-f001:**
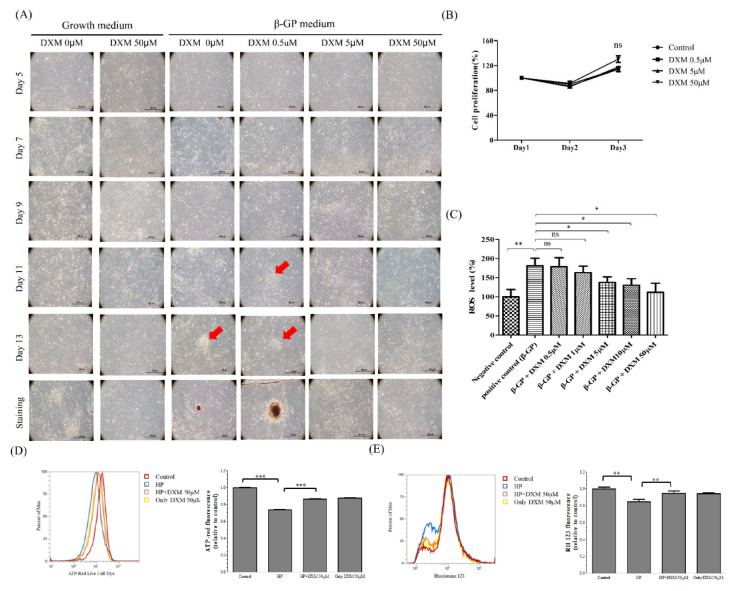
Effect of high-phosphate medium and dextromethorphan (DXM) on calcification of rat smooth vascular cells. (**A**) High-phosphate medium and DXM treatment in rat smooth vascular cells from day 5 to day 13. Cells cultured with high phosphate medium display non-specific mineral deposition (the red arrow). Alizarin red S staining for Ca deposits on day 14. (**B**) Dose-dependent effects of DXM on rat smooth vascular cell viability. (**C**) Dose-dependent effects of DXM on rat smooth vascular cells with high-phosphate medium related oxidative stress. (**D**) ATP generation and (**E**) mitochondria membrane potential (* *p* < 0.05, ** *p* < 0.01,*** *p* < 0.001, no significant difference (ns)). Abbreviations: Beta-glycerophosphate (β-GP); reactive oxygen species (ROS); high phosphate medium (HP).

**Figure 2 ijms-22-12277-f002:**
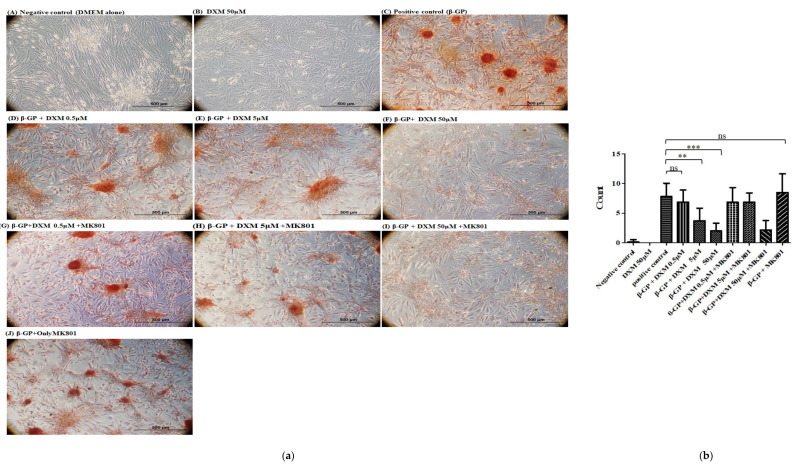
Effects of DXM plus *N*-methyl-d-aspartate receptor antagonist on vascular smooth muscle cells with the high-phosphate medium. (**a**) Alizarin red S staining was employed to assess calcification in the vascular smooth muscle cells. Scale bar, 500 μm. (**b**) Statistical analysis of calcification on each group. Not significant: ns, ** *p* < 0.01, *** *p* < 0.001.

**Figure 3 ijms-22-12277-f003:**
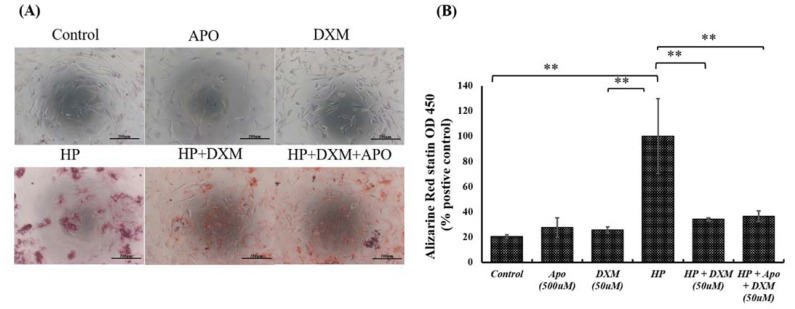
Effects of DXM on human aortic vascular smooth muscle cells with the high-phosphate medium (n = 3). (**A**) Alizarin red S staining was employed to assess calcification in the vascular smooth muscle cells. (**B**) Statistical analysis of calcification on each group. Not significant: ns, ** *p* < 0.01. Abbreviations: high-phosphate medium (HP); apocynin (APO).

**Figure 4 ijms-22-12277-f004:**
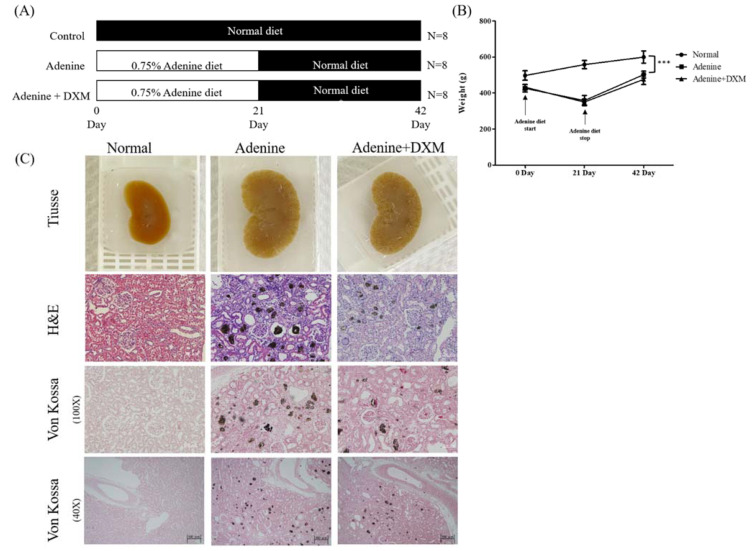
Effects of DXM in a rat model of chronic kidney disease (CKD; n = 8). (**A**) In vivo experimental design and DXM treatment schedule. (**B**) Quantification of weights of rats administered DXM orally. Not significant: ns, *** *p* < 0. 001. (**C**) Paraffin tissue sections of kidney, hematoxylin–eosin staining of kidney (magnification, ×100), and von Kossa staining of kidney (magnification, ×100 and ×40).

**Figure 5 ijms-22-12277-f005:**
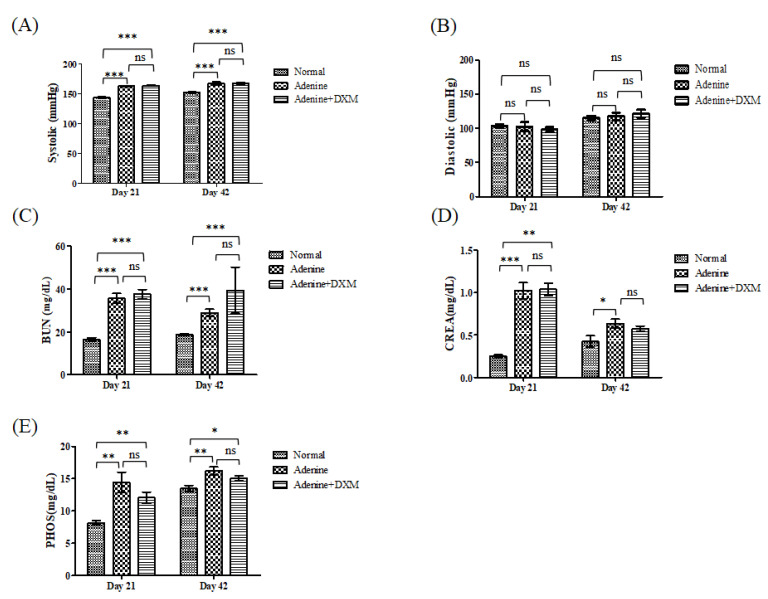
Effects of DXM on the blood pressure and biochemistry in a rat model of CKD (n = 8). (**A**) Systolic blood pressure; (**B**) diastolic blood pressure; (**C**) serum blood urea nitrogen (BUN); (**D**) creatinine (CREA); and (**E**) phosphorous (PHOS) levels during treatment. Not significant: ns, * *p* < 0.05, ** *p* < 0.01, *** *p* < 0.001.

**Figure 6 ijms-22-12277-f006:**
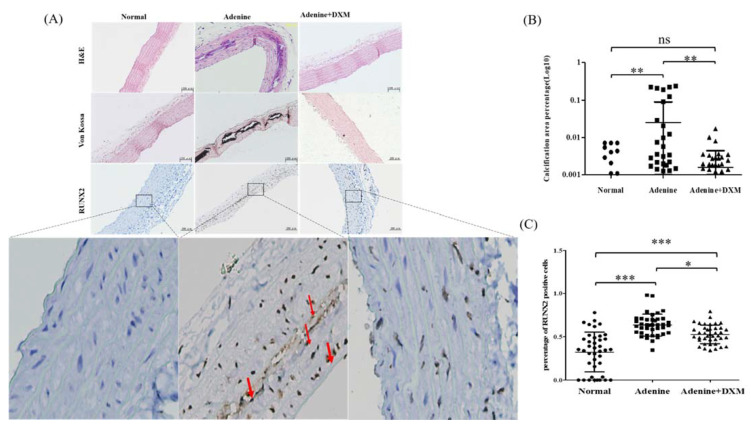
Effects of DXM on the aorta of adenine-induced chronic renal failure rats (n = 8). (**A**) Hematoxylin–eosin staining of the aortas (magnification, ×100). The von Kossa staining of aortas (magnification, ×100). Immunohistochemistry staining with antibody against runt-related transcription factor 2 (RUNX2; magnification, ×100). (**B**) Analysis of aorta calcification in each group. Not significant: ns, ** *p* < 0.01. (**C**) Analysis of percentage of RUNX2 positive staining cells in the aorta. Not significant: ns, * *p* < 0.05, ** *p* < 0.01, *** *p* < 0.001.

**Figure 7 ijms-22-12277-f007:**
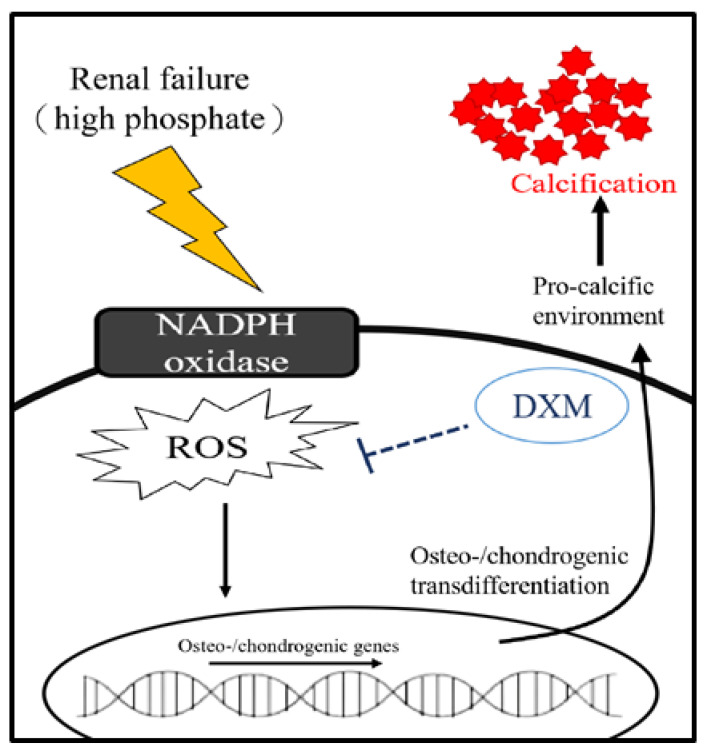
Schematic representation of the effects of dextromethorphan attenuating vascular calcification by attenuating ROS production and vascular smooth muscle cell osteoblast transdifferentiation of hyperphosphatemia.

## Data Availability

The datasets generated and/or analyzed during the current study are available from the corresponding author upon reasonable request.

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
