# Peer review of "Dextromethorphan Reduces Oxidative Stress and Inhibits Uremic Artery Calcification"

_ijms, 2021, doi:10.3390/ijms222212277_

Round 1
Reviewer 1 Report
The methods the autors used are very laborious. It was an honor for my read these medical articile.
Author Response
Reviewer 1
Comments and Suggestions for Authors
The methods the authors used are very laborious. It was an honor for my read these medical article.
Response: Thanks for your comments. The evidence relies on rigorous study designs should be clearly described in research papers. We will make the methods be clearly described in this research papers.
Reviewer 2 Report
Liu et al describe effects of dextromethorphan (DXM) on vascular calcification. To investigate this, they use VSMCs (A7r5), primary human aorta cells and an animal model (rat with kidney failure, CKD). The combination of A7r5 cells and the CKD model makes sense, both are rat. The human cells add, in my modest opinion, little or nothing. The results with these cells can be considered as a confirmation of the results in the rat A7r5 cells, but that is it. The manuscript centers around the effect of DXM on calcification in vascular cells/aorta. The problem with this manuscript is that the authors present data without a coherent mechanism. Indeed ROS (in fig.1) and RUNX2 (in fig.6) are mentioned but without a coherent line in the experimental set-up. The authors try to present some kind of a mechanism in fig.7, but that is incomplete to say the least. The discussion does not complement for the lack of a mechanism. In particular so, since it misses a clear line of reasoning.
There is a reasonable body of literature that shows the effects of mitochondrial dysfunction (by for instance calcium/phosphorous overload) on ROS generation causing vascular calcification. A high Ca/phosphorous-concentration in the medium generates excess ROS. How? DXM reduces the effect of the high phosphorous-concentration. By preventing Ca to enter the VSMCs? By restoring mitochondrial functioning?
Apparently, high Ca-concentration in the medium results in Ca-precipitation. From the figures, I am unable to deduce whether the precipitation is inside or outside the VSMCs. I suppose outside since the coloration is not clearly within the cellular boundaries. DXM reduces the precipitation. How? From the M&M section I can not deduce whether high phosphate and DXM is applied at the same time, which is important.
In the animal model adenine and phosphorous enriched diet is fed to the rats. This has, as expected, serious effects on blood pressure and serum parameters. Addition of DXM to the diet has hardly any effects on the measured parameters (fig. 4 and 5). Surprisingly, the authors find significant effects of DXM in histology (and RUNX2 measurements) (fig.6). In the discussion this discrepancy is dealt with by a single sentence (line 227). It does not make me happy.
In my opinion, the authors have described a number of interesting observations. Such observations should be followed by experimentation to find (part of) the underlying mechanisms. Or at least, focus the discussion on this item. However, the discussion provides all kind of facts on DXM which are within the scope of this study not relevant. Facts such as the antitussive properties (line 238) of DXM or that it can be bought over the counter (line 207) are, I am sorry to say, totally irrelevant. The discussion should deal with the disturbance of the mitochondrial working in relation to generation of ROS, with the link between ROS, VSMC phenotype and calcification, with the mechanisms that may be affected by DXM and so on.
In short, this potentially interesting manuscript needs more facts and more focus.
Author Response
Reviewer 2
- Liu et al describe effects of dextromethorphan (DXM) on vascular calcification.
To investigate this, they use VSMCs (A7r5), primary human aorta cells and an animal model (rat with kidney failure, CKD). The combination of A7r5 cells and the CKD model makes sense, both are rat. The human cells add, in my modest opinion, little or nothing. The results with these cells can be considered as a confirmation of the results in the rat A7r5 cells, but that is it. The manuscript centers around the effect of DXM on calcification in vascular cells/aorta. The problem with this manuscript is that the authors present data without a coherent mechanism. Indeed ROS (in fig.1) and RUNX2 (in fig.6) are mentioned but without a coherent line in the experimental set-up. The authors try to present some kind of a mechanism in fig.7, but that is incomplete to say the least. The discussion does not complement for the lack of a mechanism. In particular so, since it misses a clear line of reasoning.
Response: Thanks for your nice comments. We revised the Figure to meet the requirements about the mechanism.
(a) Our study showed that calcified medium could induce ROS. Further, DXM indeed attenuates VSMC mineralization in the study including two types of VSMCs (human and rat cells) and two types of high phosphate medium (Fig. 1-3) [11, 12]. The results may hint potential clinical practice.
(b) DXM is reported to be neuroprotective against glutamate excitatory toxicity and degen-eration of dopaminergic neurons through antagonization of the N-methyl–D-aspartic acid (NMDA) receptor. From this study, we found that the NMDA receptor antagonist could not inhibit arterial calcification. We believe that DXM decreases artery calcification independent of the NMDA receptor. Dextromethorphan was found to reduce ROS dose de-pendency in this study. Hyperphosphatemia is known to induce ROS production in Figure 1C. Oxidative stress and excessive ROS production are important mediators of osteochondrogenic transdifferentiation in VSMCs [9, 10]. Intravascular ROS could be theoretically produced by many enzymes including xanthine oxidoreductase, uncoupled nitric oxide synthase, and NADPH oxidase [24-28]. NADPH oxidase is a major source of ROS in the cardiovascular system, and are major players in mediating redox signaling under pathological conditions. NADPH oxidase inhibitors such as apocine etc. could reduce re-active oxygen species (ROS) production and block the calcified medium-induced VSMCs calcification [15, 16]. NADPH oxidase is the target of DXM action because the DXM-mediated effect disappears in NADPH oxidase-deficient mice [18]. In this experiment (Figure 3B), the protective effect of DXM was not further increased by high dose apocynin, indicating that NADPH oxidase is a major mediator of the effect of DXM. However, we cannot completely exclude other minor factors in relation to VSMC phenotype and calcification.
Evidence has demonstrated that arterial calcification is an active, cell-regulated process, based on the discovery that vascular smooth muscle cell populations are responsible for maintaining proper vascular tone and can undergo trans-differentiation into osteo-blast-like cells resulting in increased vascular stiffness [4-8]. Our report showed that calci-fication was significantly reduced in aortic area when dextromethorphan was added. The effects of dextromethorphan on the master regulator of vascular bone protein (RUNX 2) were also decreased compatible with vascular calcification-Von Kossa staining results. Elevated serum phosphorus levels are known to promote vascular calcification in patients with CKD [11-14, 29]. In this study, phosphorus, urea, and creatinine levels were elevated in rats, but treatment with dextromethorphan had no significant impact on the renal func-tion of adenine-induced renal failure. There were no significant changes in blood pressure between adenine-fed rats with and without dextromethorphan, although DXM was con-firmed to improve endothelial function, decreased blood pressure [20, 21] and reduce the thickness of the medial layer of the aorta in hypertensive rats [19]. This discrepancy be-tween the no change of blood pressure and its antioxidant effects on NADPH oxidase is hard to be explained. Rats fed adenine showed the typical features of CKD that included elevation of blood pressure, decreased growth, increased water intake and decrease in cre-atinine clearance. There could be some other factors such as fluid status that may offset the beneficial effects of DXM such as lowering blood pressure. However, this indicates that the action of dextromethorphan against vascular calcification is independent of renal failure improvement and mechanical stretch by blood pressure changes. The anti-vascular calcification of DXM mostly depends on its direct action on local vascular smooth muscle cells because of its antioxidant effects on NADPH oxidase.
(c) RUNX2 (in fig.6), H.E. staining and Von-Kossa staining confirmed the effects of DXM to inhibit vascular calcification.
- There is a reasonable body of literature that shows the effects of mitochondrial
dysfunction (by for instance calcium/phosphorous overload) on ROS generation causing vascular calcification. A high Ca/phosphorous-concentration in the medium generates excess ROS. How? DXM reduces the effect of the high phosphorous-concentration. By preventing Ca to enter the VSMCs? By restoring mitochondrial functioning?
Response:
The ability of DXM to block NMDA receptors would unlikely be responsible for the protective effect of DM, because blockers of NMDA receptors- MK801 blocker could not produce the similar effects. NADPH oxidase is a major source of ROS in the cardiovascular system, and are major players in mediating redox signaling under pathological conditions. NADPH oxidase is the target of DXM action because the DXM-mediated effect disappears in NADPH oxidase-deficient mice [18]. NADPH oxidase inhibitors such as apocynin etc. could reduce reactive oxygen species (ROS) production and block the calcified medium-induced VSMCs calcification [15, 16]. In this experiment (Figure 3B), the protective effect of DXM was not further increased by high dose apocynin, indicating that NADPH oxidase is a major mediator of the effect of DXM. However, we cannot completely exclude other minor factors in relation to VSMC phenotype and calcification.
- Apparently, high Ca-concentration in the medium results in Ca-precipitation.
From the figures, I am unable to deduce whether the precipitation is inside or outside the VSMCs. I suppose outside since the coloration is not clearly within the cellular boundaries. DXM reduces the precipitation. How? From the M&M section I can not deduce whether high phosphate and DXM is applied at the same time, which is important.
Response:
Thanks for your comments. High phosphate and DXM applied at the same time was described in the methods.
- In the animal model adenine and phosphorous enriched diet is fed to the rats. This
has, as expected, serious effects on blood pressure and serum parameters. Addition of DXM to the diet has hardly any effects on the measured parameters (fig. 4 and 5). Surprisingly, the authors find significant effects of DXM in histology (and RUNX2 measurements) (fig.6). In the discussion this discrepancy is dealt with by a single sentence (line 227). It does not make me happy.
Response: Thanks for your nice comments.
This discrepancy between the no change of blood pressure and its antioxidant effects on NADPH oxidase is hard to be explained. Rats fed adenine showed the typical features of CKD that included elevation of blood pressure, decreased growth, increased water intake, and decrease in creatinine clearance. There could be some other factors such as fluid status that may offset the beneficial effects of DXM such as lowering blood pressure. However, this indicates that the action of dextromethorphan against vascular calcification is independent of renal failure improvement and mechanical stretch by blood pressure changes. The anti-vascular calcification of DXM mostly depends on its direct action on local vascular smooth muscle cells because of its antioxidant effects on NADPH oxidase.
In my opinion, the authors have described several interesting observations.
Such observations should be followed by experimentation to find (part of) the underlying mechanisms. Or at least, focus the discussion on this item. However, the discussion provides all kind of facts on DXM which are within the scope of this study not relevant. Facts such as the antitussive properties (line 238) of DXM or that it can be bought over the counter (line 207) are, I am sorry to say, totally irrelevant. The discussion should deal with the disturbance of the mitochondrial working in relation to generation of ROS, with the link between ROS, VSMC phenotype and calcification, with the mechanisms that may be affected by DXM and so on.
In short, this potentially interesting manuscript needs more facts and more focus.
Response: Thanks for your comments.
**the discussion provides all kind of facts on DXM which are within the scope of this study not relevant. Facts such as the antitussive properties (line 238) of DXM or that it can be bought over the counter (line 207) are, I am sorry to say, totally irrelevant.
**
This means that DXM has a very impressive clinical safety record because it was widely used as an over-the-counter anti-cough agent for several decades. This indicated that dextromethorphan can be a potential clinical drug for preventing vascular calcification in patients with CKD.
In this experiment (Figure 3B), the protective effect of DXM was not further increased by high dose apocynin, indicating that NADPH oxidase is a major mediator of the effect of DXM. However, we cannot completely exclude other minor factors in relation to VSMC phenotype and calcification.
Reviewer 3 Report
This study is generally well-written and results are of interest.
Relevant ethical disclosures and handling of animals were appropriate and disclosed where appropriate.
Statistical methods used are valid.
Results seem to be sound,
I only have some minor objections, as outlined below:
- The abstract should be rewritten completely. Authors should cut the Background part of the Abstract significantly and expand on the Results section where some numerical main results should be reported. In this current way, the Abstract is vague.
- Please fix the subtitle under heading 3.4
- Please provide Figure 7 in color.
- In the Discussion, please replace "several evidences" with singular "evidence"
- The translational perspective on dextromethorphan in potential clinical use should be briefly elaborated in the Discussion section.
Author Response
Reviewer 3:
This study is generally well-written and results are of interest.
Relevant ethical disclosures and handling of animals were appropriate and disclosed where appropriate.
Statistical methods used are valid.
Results seem to be sound,
I only have some minor objections, as outlined below:
1.The abstract should be rewritten completely. Authors should cut the Background part of the Abstract significantly and expand on the Results section where some numerical main results should be reported. In this current way, the Abstract is vague.
Response: The abstract has been rewritten completely.
- Please fix the subtitle under heading 3.4
Response: The subtitle has been revised under heading 3.4.
3.Please provide Figure 7 in color.
Response: The Figure 7 has been in color.
- In the Discussion, please replace "several evidences" with singular "evidence"
The translational perspective on dextromethorphan in potential clinical use should be briefly elaborated in the Discussion section.
Response:
- The discussion, several evidence has been corrected with singular evidence.
Evidence has demonstrated that arterial calcification is an active, cell-regulated process, based on the discovery that vascular smooth muscle cell populations are responsible for maintaining proper vascular tone and can undergo trans-differentiation into osteoblast-like cells resulting in increased vascular stiffness [4-7].
- The translational perspective on dextromethorphan in potential clinical use should
be briefly elaborated in the Discussion section.
Dextromethorphan, an over-the-counter antitussive agent, is one of the most widely used active ingredients as a cough suppressant among cold and cough medications, with high safety and efficacy at recommended doses. Dextromethorphan is an effective antitussive agent that is rapidly absorbed when administered orally and is excreted mostly through the urine. It has largely replaced codeine as a cough suppressant. It is devoid of analgesic properties unlike codeine. Compared to codeine, Dextromethorphan produces less respiratory depression, less gastrointestinal disturbances, and less drug dependence or abuse. Dextromethorphan has been safely administered orally at 10–40 mg/kg in mice [19]. DXM reduces oxidative stress and inhibits the typical inflammatory disease with in-volved macrophage proliferation and intimal artery calcification (atherosclerosis) in mice [19]. Addiction does not usually occur, even after large doses for prolonged periods [19, 29, 30]. At equipotent doses for local anesthesia, DXM was found to be safer than bupivacaine (a long-acting local anesthesia) in the central nervous system and in cardiovascular toxicity. The highest subcutaneous injected dose of DXM was up to 20 µmoL/kg [31]. Accordingly, DXM orally at doses of 20 mg/kg/day in this study on rats were safe for vascular calcification actions including intimal artery calcification (atherosclerosis) and medial artery calcification (arteriosclerosis). DXM has a very impressive clinical safety record because it was widely used as an over-the-counter anti-cough agent for several decades. Due to its proven safety record of long-term clinical use in humans, DXM may be a therapeutic strategy for targeting vascular calcification and atherosclerosis in CKD, and provides strong rationale for further studies.
Round 2
Reviewer 2 Report
In responds on the reviewer’s comments, Liu et al have made some changes/additions in the text. In my original review I have indicated that the original manuscript does not answer a number of obvious and pressing questions. The new text does not answer such questions. I have suggested additional experimental work. Unfortunately, the authors have chosen not to do so. Thus, this manuscript contains an interesting observation, but that is it.
The authors themselves, indicate that the manuscript is far from complete. In lines 249/250 they state: “This discrepancy…. explained”. In my opinion, this remark, and other less explicit ones, show that the authors realize that some mechanistic proof, or at least hypothesis has to be presented. This is underlined by line 280: “The present study…. to be addressed. First, …..” I am missing ‘second’ and ‘third’ and even more. Second, for instance, lines 126-156 describe the (lack of) effect of DXM on the physiology of adenine treated rats. A hidden remark can be found in line 155/156. This lack of physiological effect coincides with significant effects on calcification and RUNX2. This needs some kind of explanation!! Third, if NMAD receptor can be excluded to be involved in DXM action (line 104), then which signaling pathway is involved? And so on.
Most disturbing is the lack of a link between the experiments on cultured cells and the animal model CKD. Cells can be used to detect which signal pathways may be involved. Candidate genes may then be knocked out/modified in experimental animals to see whether still affect the calcification in the far more complex setting of a blood vessel (including endothelial cells, adventitial cells and so on). For instance, in this manuscript NMAD is discarded based on the outcome of experiments with cultured cells, to, out of the blue, present RUNX2 for the animal experiments. Why not Sox9 or HIF1?
Once again, this manuscript needs more than the observation that DXM does something to vascular calcification. The authors should seriously try to come up with a more coherent line of experimentation and reasoning.
Small stuff:
- A considerable number of typing errors, even in the reference section.
- The text is here and there rather repetitive. Particular disturbing is the identical sentence in lines 61/62 and 64/65. This gives the impression that the authors have used copy-paste writing.
- What kind of molecule is MK801? (line 306)
Author Response
Reviewer 2
In response on the reviewer’s comments, Liu et al have made some changes/additions in the text. In my original review, I have indicated that the original manuscript does not answer a number of obvious and pressing questions. The new text does not answer such questions. I have suggested additional experimental work. Unfortunately, the authors have chosen not to do so. Thus, this manuscript contains an interesting observation, but that is it.
The authors themselves, indicate that the manuscript is far from complete. In lines 249/250 they state: “This discrepancy…. explained”. In my opinion, this remark, and other less explicit ones, show that the authors realize that some mechanistic proof, or at least hypothesis has to be presented. This is underlined by line 280: “The present study…. to be addressed. First, …..” I am missing ‘second’ and ‘third’ and even more. Second, for instance, lines 126-156 describe the (lack of) effect of DXM on the physiology of adenine treated rats. A hidden remark can be found in line 155/156. This lack of physiological effect coincides with significant effects on calcification and RUNX2. This needs some kind of explanation!! Third, if NMAD receptor can be excluded to be involved in DXM action (line 104), then which signaling pathway is involved? And so on.
Most disturbing is the lack of a link between the experiments on cultured cells and the animal model CKD. Cells can be used to detect which signal pathways may be involved. Candidate genes may then be knocked out/modified in experimental animals to see whether still affect the calcification in the far more complex setting of a blood vessel (including endothelial cells, adventitial cells and so on). For instance, in this manuscript NMAD is discarded based on the outcome of experiments with cultured cells, to, out of the blue, present RUNX2 for the animal experiments. Why not Sox9 or HIF1?
Once again, this manuscript needs more than the observation that DXM does something to vascular calcification. The authors should seriously try to come up with a more coherent line of experimentation and reasoning.
Answer: Your comments could make the paper better. Thanks anyway for comments.
The topic of the study is ** Dextromethorphan Reduces Oxidative Stress and Inhibits Uremic Artery Calcification**
Dextromethorphan (DXM), an over-the-counter antitussive drug, is one of the most widely used active ingredients as a cough suppressant in cold and cough medications. DXM is an antagonization of the N-methyl–d-aspartic acid (NMDA) receptor and NADPH oxidase inhibitor since DXM may effectively inhibit the production of reactive oxygen species (ROS) induced by 1-methyl-4-phenyl-1,2,3,6-tetrahydropyridine. However, it was not known whether DXM may provide additional cardiovascular protection to renal failure patients. We used in vitro and in vivo studies to evaluate the effect of DXM on artery changes in the presence of hyperphosphatemia. The anti-vascular calcification effect of DXM was tested in adenine-fed Wistar rats. High-phosphate medium induced ROS production and calcification of VSMCs. DXM significantly attenuated the increase in ROS production, the decrease in ATP and mitochondria membrane potential during calcified-medium–induced VSMC calcification process (p < 0.05). The protective effect of DXM in calcified-medium–induced VSMC calcification was not further increased by NADPH oxidase inhibitors, indicating that NADPH oxidase mediates the effect of DXM. Furthermore, DXM decreased aortic calcification in Wistar rats with CKD. Our results suggest that treatment with DXM can attenuate vascular oxidative stress and ameliorate vascular calcification.
Question 1:
“This discrepancy…. explained”. In my opinion, this remark, and other less explicit ones, show that the authors realize that some mechanistic proof, or at least hypothesis has to be presented.
Answers: Thanks anyway for the comments.
There were no significant changes in blood pressure between rats fed the adenine diet with and without DXM, although DXM improved endothelial function, decreased blood pressure [20, 21], and reduced the thickness of the medial layer of the aorta in hypertensive rats [19]. This discrepancy between the lack of change in blood pressure and its antioxidative effects on NADPH oxidase is difficult to explain. The possible cause is the DXM dosages. The effect of DXM on blood pressure had been studied and the result showed that such effects are not dose-dependent. A low dose rather than a high dose of DXM could reduce blood pressure in experimental hypertension [20]. The extremely higher dose of DXM as the NMDA antagonist produces serotonergic-glutamatergic interactions in the mechanism of action of classic hallucinogens and increased blood pressure and heart rate [31] Hence, the interaction may offset the beneficial effects of DXM, such as lowering blood pressure. Daily oral DXM 20 mg/kg (Sigma-Aldrich) may be a higher dose of DXM for rats with CKD in concern of blood pressure. Future investigations are required to define the optimal dose of DM before it could be used for another treatment dose in animals with CKD and patients with CKD [20]. Taken together, this study indicates that the action of DXM against vascular calcification is independent of renal-failure improvement and mechanical stretch due to blood pressure changes. The anti-vascular calcification of DXM mostly depends on its direct action on local VSMCs because of its antioxidative effects.
Question 2:
This is underlined by line 280: “The present study…. to be addressed. First, …..” I am missing ‘second’ and ‘third’ and even more. Second, for instance, lines 126-156 describe the (lack of) effect of DXM on the physiology of adenine treated rats. A hidden remark can be found in line 155/156. This lack of physiological effect coincides with significant effects on calcification and RUNX2. This needs some kind of explanation!! Third, if NMAD receptor can be excluded to be involved in DXM action (line 104), then which signaling pathway is involved? And so on.
Answers: Thanks anyway for the comments.
The present study has some limitations that need to be addressed. First, the results of our study could not be directly extrapolated to humans, as data on other animal models are limited; therefore, further investigations are needed in other models or humans. However, the findings of the current study provide evidence for the involvement of oxidative stress in the mechanism underlying vascular calcification and indicate the potential for DXM as a clinical drug for preventing vascular calcification in patients with CKD. Second, this study confirmed the effects of dextromethorphan (DXM), on oxidative stress and vascular calcification. Actually, artery calcification is the far more complex set of a blood vessel (including endothelial cells, adventitial cells, and so on). We only showed the vascular calcification evidence by hematoxylin-eosin staining, von-Kossa staining, and osteogenic differentiation of VSMC by RUNX2 in animal studies. DXM significantly attenuated the increase in ROS production, the decrease in ATP and mitochondria membrane potential during calcified-medium–induced VSMC calcification process (p < 0.05). Further detail mechanism needs further studies. However, we cannot completely exclude other signaling pathway effects.
Question 3:
Why not Sox9 or HIF1?
Answers: Thanks anyway for the comments.
Hypoxia contributes to vascular calcification through the induction of osteochondrogenic differentiation of VSMCs in an HIF-1–dependent. The transcription factor Sox9 is expressed in all chondroprogenitors and has an essential role in chondrogenesis
Sox9 or HIF1 are associated with vascular smooth muscle cell transdifferentiation, but not common osteogenic markers. The markers of vascular smooth muscle cells osteogenic transdifferentiation are usually referred to, RUNX2 (runt-related transcription factor 2), OCN (osteocalcin), or ALP (alkaline phosphatase). We can not do all markers, so I choose RUNX2.
Question 4:
A considerable number of typing errors, even in the reference section. The text is here and there rather repetitive. Particular disturbing is the identical sentence in lines 61/62 and 64/65. This gives the impression that the authors have used copy-paste writing
Answer: we have revised it.
Question 5: What kind of molecule is MK801? (line 306)
Answer: an NMDA receptor antagonist (dizocilpine, MK-801, Sigma-Aldrich)

Round 3
Reviewer 2 Report
The authors have added one experiment and rewritten a few sections. The experiments do not shed light on the mechanisms of DXM action, neither does is improve the relation between cell culture work and animal model. The most important textual improvement is the addition of a remark that this study has some limitations (lines 281-295). But then (line 283) the authors claim to “provide evidence for the involvement of oxidative stress in vascular calcification” as have hundreds of articles before. NEW is the possible effect of DXM on the calcification process. However, the data on DXM are difficult to interpret, firstly because of the use of two very different systems, secondly because the complexity of the adenine-induced kidney failure animal model
The major question remains: how to connect the in vitro assays with the animal model? To put it in a nutshell: in vitro high phosphate induces ‘calcification’ which can be reduced by DXM; in vivo, adenine caused amongst many other effects, high phosphate serum levels, resulting in calcification. However, DXM does not reduce serum phosphate levels although it prevents/repairs calcification. So, the question arises what high phosphate levels have to do with calcification and consequently what is the mechanism behind the action of DXM?
In that respect it is worthwhile to look at an article of a Swedish group (not referred to in this manuscript), of which I have copied part of the abstract:
Ambulatory systolic and diastolic blood pressures measured by radiotelemetry were significantly elevated in A-CRF (adenine-induced chronic failure) animals from week 3and onward. At death, A-CRF animals had anemia, hyperphosphatemia, hyperparathyroidism, and elevated plasma levels of asymmetric dimethylarginine and oxidative stress markers. There were no significant differences between groups in the sensitivity, or maximal response, to ACh, sodium nitroprusside (SNP), norepinephrine, or phenylephrine in either mesenteric arteries or aortas. However, in A-CRF animals, the rate of aortic relaxation was significantly reduced following washout of KCl (both in intact and endothelium-denuded aorta) and in response to ACh and SNP. Also the rate of contraction in response to KCl was significantly reduced in A-CRF animals both in mesenteric arteries and aortas. The media of A-CRF aortas was thickened and showed focal areas of fragmented elastic lamellae and disorganized smooth muscle cells. No vascular calcifications could be detected. These results indicate that severe renal failure for a duration of less than 10 wk in this model primarily affects the aorta and mainly slows the rate of relaxation. Nguy-L, Nilsson-H, Lundgren-J et al. 2012.
General: What does adenine do to cultured vascular cells? Probably nothing. However, when certain concentrations of adenine circulate in the animal body, vascular cells will be exposed and probably affected and therefore representative concentrations of adenine should be tested on cultured vascular cells if one wants to have both cells and animals in one study.
Do calcium depositions in cell culture have the same chemical composition compared with the vascular deposits?
What is the relation between mitochondria (ATP production) and ROS generation?
Fig 1: D, E: ATP is largely generated in mitochondria. Thus, high phosphate medium reduces mitochondrial membrane potential and consequently ATP generation, but ROS production is doubled. DXM ameliorates the effect of high phosphate, but ….. it also reduces the production of ATP and of ROS. Additionally, although I do not have the figures hidden behind the graphs, the indicated standard deviation makes me believe that the reduction of membrane potential (E) and ATP (D) by DXM is significant whether high phosphate or not. So, the effect of DXM on the mitochondria of these cells has nothing to do with the phosphate concentration.
In Fig 1A why not ‘day 14’ instead of the meaningless ‘staining’?
Fig 2: What is the negative and positive control? In the M&M are no phosphate concentrations mentioned (line 302-308).
Fig 3: Where is the HP+Apo group?
Fig 4 and 5: DXM has no effect on any of the reported blood parameters. SO, once again, what is the mechanism behind the DXM action. Adenine has an effect but that is not what this study is about.
Fig 6: Calcification areas of the aorta of the adenine-treated animals are for most animals within the range of the control group (60%), with only 30% with significantly more calcification. Which rat in the graph is the picture of!!!? Besides, the logarithmic scale is somewhat deceptive. So, how representative are the pictures in Fig.6? Interesting are the large difference between the individual rats. Can it be correlated with some physical or hematological parameter? Some aspects of the aorta on the picture are reminiscent to 'aortic dissection".
In the same figure the immune-histochemical staining against RUNX2 is presented. The positive cells (red arrows, I suppose) are hard to see. Additionally, the blue background staining in control and adenine+DXM sections is more prominent than in the adenine section.
Fig 7: According to this scheme ROS concentration is increased by high phosphate. However, the in vitro experiments show a decrease of ATP production under high phosphate conditions. These results appear to be conflicting. Additionally, the results of the blood analysis and the calcification as observed in sections are conflicting and hardly indicate the role of DXM as presented in this figure. In particular the relation between NADPH and DXM (lines 113/114) where NADPH is presented as mediator for DXM.